# Mycobiome Study Reveals Different Pathogens of Vulvovaginal Candidiasis Shape Characteristic Vaginal Bacteriome

Changying Zhao,[a,b,d] Ying Li,[c] Bin Chen,[a,b] Kaile Yue,[a,b] Zhenzhen Su,[a,b] Jing Xu,[c] Wanhua Xue,[c] Guoping Zhao,[a,b,e,f] (ORCID) Lei Zhang[a,b,d,e]

[a]Department of Biostatistics, School of Public Health, Cheeloo College of Medicine, Shandong University, Jinan, China
[b]Microbiome-X, National Institute of Health Data Science of China, Cheeloo College of Medicine, Shandong University, Jinan, China
[c]Qilu Hospital of Shandong University Dezhou Hospital & Dezhou People's Hospital, Dezhou, China
[d]Shandong Children's Microbiome Center, Children's Hospital affiliated with Shandong University, Jinan, China
[e]State Key Laboratory of Microbial Technology, Shandong University, Qingdao, China
[f]CAS Key Laboratory of Computational Biology, Bio-Med Big Data Center, Shanghai Institute of Nutrition and Health, University of Chinese Academy of Sciences, Chinese Academy of Sciences, Shanghai, China

Changying Zhao, Ying Li, and Bin Chen contributed equally to this article. Author order was determined by the corresponding authors and co-authors.

**ABSTRACT** Vulvovaginal candidiasis (VVC) can alter the vaginal microbiome composition and structure, and this may be correlated with its variable treatment efficacy. Integrated analysis of the mycobiome and bacteriome in VVC could facilitate accurate diagnosis of infected patients and further decipher the characterized bacteriome in different types of VVC. Our mycobiome analysis determined two common types of VVC, which were clustered into two community state types (CSTs) featured by *Candida glabrata* (CST I) and *Candida albicans* (CST II). Subsequently, we compared the vaginal bacteriome in two CSTs of VVC and two other types of reproductive tract infections (RTIs), bacterial vaginosis (BV) and *Ureaplasma urealyticum* (UU) infection. The vaginal bacteriome in VVC patients was between the healthy and other RTIs (BV and UU) status, it bore the greatest resemblance to that of healthy subjects. While BV and UU patients have the unique vaginal microbiota community structure, which very different with healthy women. Compared with CST II, the vaginal bacteriome of CST I VVC was characterized by *Prevotella,* a key signature in BV. In comparison, CST II was featured by *Ureaplasma*, the pathogen of UU. The findings of our study highlight the need for co-analysis and simultaneous consideration of vaginal mycobiome and bacteriome in the diagnosis and treatment of VVC to solve common clinical problems, such as unsatisfactory cure rates and recurrent symptoms.

**IMPORTANCE** Fungi headed by *C. albicans* play a critical role in VVC but are not sufficient for its occurrence, indicating the involvement of other factors, such as the vaginal bacteriome. We found that different CST correspond to different bacterial composition in patients with VVC, and this could underlie the alteration of vaginal microorganism environment in VVC patients. We believe that this correlation should not be ignored, and it may be related to the unsatisfactory treatment outcomes and high recurrence rate of VVC. Here, we provided evidence for associations between vaginal bacteriome patterns and fungal infection. Screening specific biomarkers for three common RTIs paves a theoretical basis for further development of personalized precision treatment.

**KEYWORDS** vulvovaginal candidiasis, candidiasis, mycobiome, fungal community, RTIs

Address correspondence to Lei Zhang, zhanglei7@sdu.edu.cn, or Guoping Zhao, guopingzhao@yeah.net.

The authors declare no conflict of interest.

Vulvovaginal candidiasis (VVC) is a widespread vaginal infection primarily caused by *Candida* and affecting 75% of the women of childbearing age (1, 2). The first-line treatments for VVC are usually empirical and performed with azoles or polyene drugs, which have shown increasing ineffectiveness (3). Several studies reveal that the relative abundance of vaginal microbes changed during VVC, such as *Lactobacillus* decreased

10.1128/spectrum.03152-22 1

while *Prevotella*, *Sneathia*, and *Atopobium* increased. All these changes in the vaginal environment are generally required for the alteration of the opportunistic *Candida* species from commensal to pathogenic (4, 5). The ultimate goal of treatment of vaginal infections is to recover the normal vaginal microecology disrupted by vaginal infections and prevent recurrences (6). Bacteria have long been known to dominate the vaginal milieu, leading to a number of studies about vaginal microecology that have focused only on the bacterial community (7). However, exploring the composition of fungal species and fungal-bacterial association in the VVC vaginal environment is also important (8).

In the current clinical setting, the diagnosis of vulvovaginal candidiasis is made using a combination of clinical signs and symptoms with potassium hydroxide microscopy (9). Culture-dependent techniques could recover vaginal mycobiota, in which *C. albicans* predominate (72% to 91%) (10). The cultivation-dependent method, however, is limited by the use of culture media for multiple microbes, and some microorganisms from the vaginal tract are difficult to cultivate because of nutritional and anaerobic requirements (11). Hence, the diversity of vaginal mycobiota cannot be fully appreciated in clinical laboratories. It has been suggested that the widespread and inappropriate use of antifungal treatments may lead to the selection of non-*Candida albicans Candida* (NCAC) species (such as *C. glabrata*), which are more resistant to the commonly used antifungal agents than *C. albicans* (5). This scenario indicated the need for further studies based on the mycobiome of VVC pathogens to improve the existing understanding of VVC pathogenesis.

Reports on the relationship between fungal colonization and vaginal bacteria have been contentious. For example, the decline of *Lactobacilli* predominance and the overgrowth of opportunistic pathogens may lead to occurrence of VVC (12), whereas *Lactobacilli* can establish commensalism with *Candida*, producing lactate to make *Candida* less vulnerable to immune responses (13, 14). Although the vaginal microbiota in VVC is not as clear as that in bacterial vaginosis, *Candida* and bacteria both play a major role in sustaining *Candida* commensal form through bacteria-fungal interactions (15). The changes of the bacterial communities occurring during the VVC could result in significant alterations in the vaginal metabolites composition, which may be further associated with other RTIs (4, 16).

However, the effects of common types of VVC pathogens on their corresponding vaginal bacteriome remains unknown. This suggests that exploring the types of fungal species in parallel with bacteriome in the context of the vaginal environment is important, with potential implications for treating and preventing VVC, improving obstetric outcomes and reproductive health in general (8). In the present study, we aimed to decipher the vaginal microorganism environment, including bacteria and fungi in different types of VVC women, and identify the bacteriome features across three common RTIs (VVC, BV and UU), in order to provide new insights into the etiology, treatment and amalgamative infections of VVC.

## RESULTS

**Study population.** For the characterization of the vaginal microbiota associated with VVC, we recruited 114 RTI subjects (including 37 BV patients, 44 VVC patients, and 33 UU patients) and 47 healthy controls. We conducted fungal internal transcribed spacer (ITS) gene sequencing to analyze VVC vaginal swabs samples and used bacteria 16S rRNA gene sequencing to analyze those 161 vaginal swabs samples. The clinical characteristics of subjects are summarized in Table S1. No significant differences were noticed in either age or gender between RTIs and healthy control groups.

**Vulvovaginal candidiasis is mainly caused by *C. albicans* and *C. glabrata* infections.** The internal transcribed spacer (ITS) of fungal genes from vaginal samples collected from VVC subjects were amplified. Two community state types (CST) were identified and found to be driven by a relatively high abundance of the species *C. glabrata* (CST I, *n* = 9) and *C. albicans* (CST II, *n* = 35), using an unsupervised cluster analysis (Fig. 1A).The analysis of beta diversity revealed that the vaginal mycobiota of CST I was different compared to CST II as calculated by principal coordinates analysis (PCoA) and PERMANOVA on the unweighted UniFrac distance ($R^2$ = 0.072; *P* = 0.002; Fig. 1B).

**The interaction of fungus and bacteria in the vaginal of VVC patients.** To elucidate the potential bacterial-fungal interactions in the vaginal of the VVC patients, we next

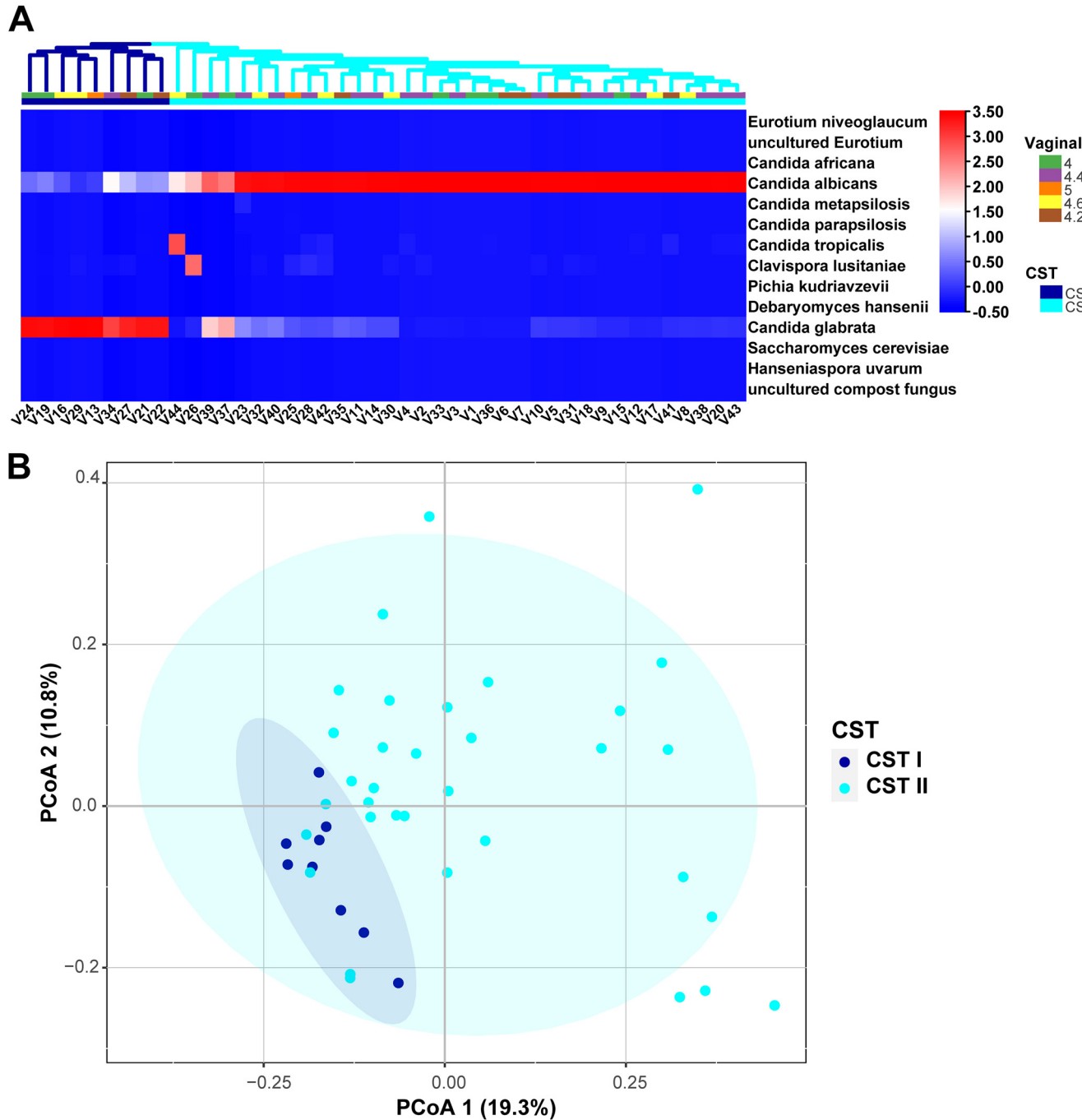

**FIG 1** Vulvovaginal candidiasis is mainly caused by *C. albicans* and *C. glabrata* infections. (A) Unsupervised hierarchical clustering of fungal for VVC. *C. glabrata* was dominant in CSTI, and *C. albicans* was dominant in CSTII. (B) PCoA of fungi community beta diversity based on the unweighted UniFrac distances between CST I and II.

analyzed the correlations between the relative abundance of bacterial and fungal taxa at the genus and species levels, respectively (Fig. 2A). We found that several bacteria show inconsistent relationship with *C. albicans* and *C. glabrata*. For instance, *Acidaminococcus* has statistically significant positive correlations with the *C. albicans*, while it has significant negative correlations with the *C. glabrata*. Furthermore, bacteria were assessed in VVC vaginal samples, to determine whether they were associated with the classification of CST I and CST II. Linear discriminant effect size (LEfSe) analysis identified *Ureaplasma*, the pathogen of UU, and *Bifidobacterium adolescentis* exhibited higher abundances in CST II. While we also found that the colonization of *C. glabrata* was positively correlated with BV

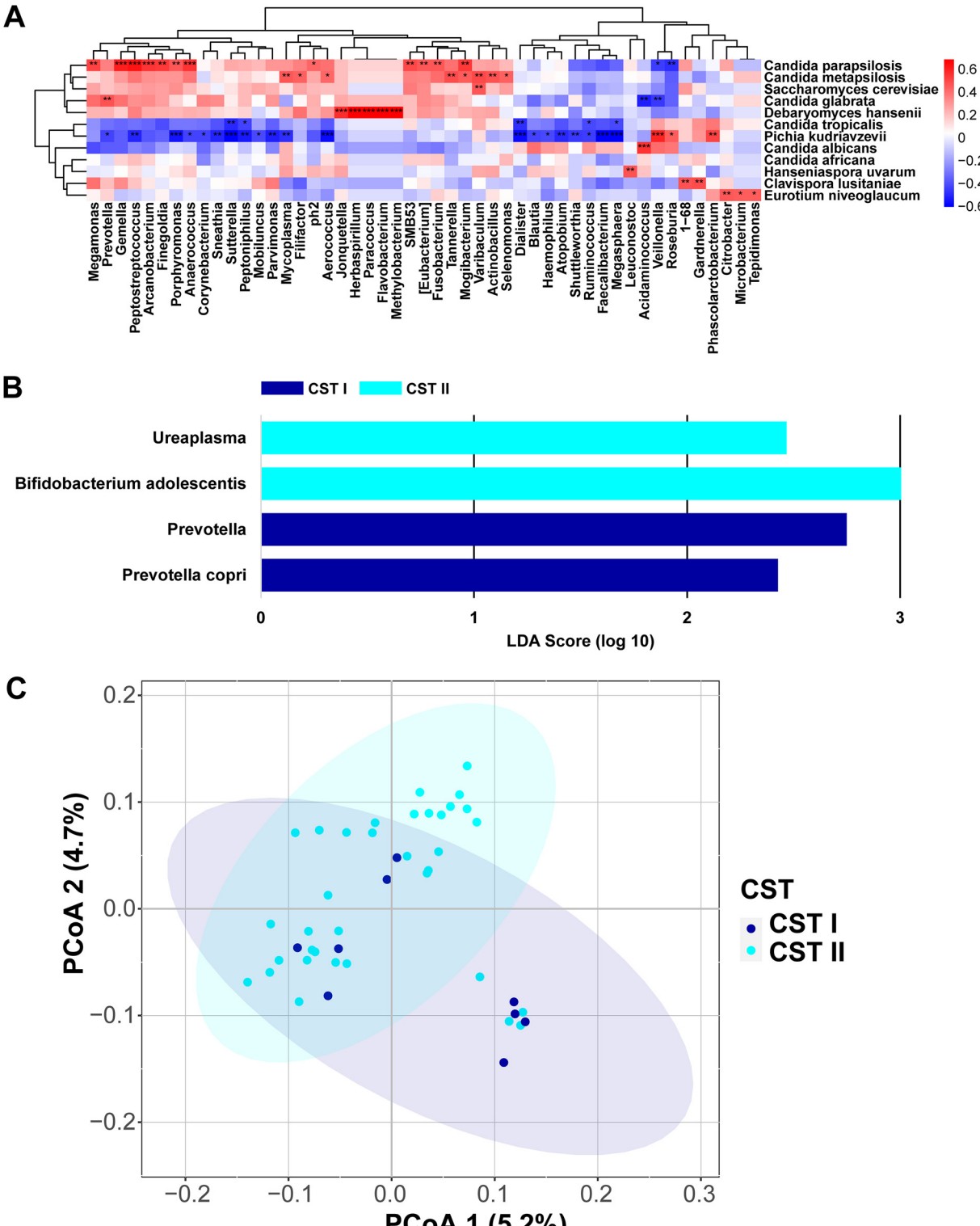

**FIG 2** Interaction between vaginal fungi and bacteria during vulvovaginal candidiasis. (A) The heatmap depict Spearman correlation of bacterial and fungal. The R values are represented by gradient colors, where red and blue indicate positive and negative correlations, respectively (*, $P < 0.05$; **, $P < 0.01$; ***, $P < 0.001$). (B) Histogram of the LDA scores computed for different abundance levels between CST I and II. (C) PCoA of bacterial beta diversity analysis based on unweighted UniFrac distance between CST I and II.

common opportunistic bacteria, including *Prevotella* genus (Fig. 2B). The composition of vaginal bacterial community was different between CST I and CST II analysis of beta diversity based on the unweighted UniFrac (PERMANOVA, $R^2 = 0.027$, $P = 0.037$, Fig. 2C), suggesting that fungal indeed could play a role in the altered microbial community associated with the VVC subjects.

We implemented BugBase using default parameters to predict CST I and CST II microbial phenotypic characteristics using 16S rRNA gene sequence data (Fig. S1). BugBase determined the proportion of each bacterial community sample, including Gram-positive, Gram-negative, biofilm-forming, pathogenic potential, mobile element containing, oxygen utilizing and oxidative stress tolerant. We selected vaginal pH and 5 organism-level microbial phenotypic characteristics of interest for further analysis with the Wilcoxon test. No significant difference was found between CST I and CST II. Overall, the bacterial and fungal interactions in the vagina are complex and undefined; further investigations remain to be done.

**Fungal infections are closer to the level of the healthy group bacterial communities compared to BV and UU.** To further describe the stability and dynamics of the vaginal bacterial community of VVC patients, at the bacterial level, in addition to the healthy control group, we included the vaginal bacterial community of BV and UU patients into the analysis. Firmicutes, Bacteroidetes and Actinomycetes were the main bacteria groups in the reproductive tract of healthy women (Fig. 3C). Phylum level analysis showed a clear alteration of the bacterial vaginal community in RTIs characterized by a lower Firmicutes/Bacteroidetes ratio in the RTI group compared to the control group due to a significant reduction of the relative abundance of Firmicutes. Genus level analysis showed that the BV group has the largest difference in flora structure compared with healthy women. In the BV group, the relative abundance of *Lactobacillus* decreased to about 15%, while the relative abundance of *Prevotella*, *Sneathia*, and *Atopobium* increased significantly compared with healthy women. In the UU group, the abundance of *Lactobacillus* decreased to less than 60%, while the abundance of *Bifidobacterium* and *Streptococcus* increased. Compared with the healthy control group, the bacterial community structure of the VVC group changed the least. The abundance of *Lactobacillus* decreased to less than 80%, while the abundance of *Bifidobacterium* increased (Fig. 3C). Analysis of alpha diversity revealed a significant increase in microbial diversity and richness ($P < 0.05$, Fig. 3A) in the three RTI groups, compared with healthy control. Among the three RTI groups, the BV group had the highest vaginal bacterial flora richness, followed by the UU group and the VVC group. The analysis of the beta diversity calculated on the unweighted UniFrac distances dissimilarity revealed that the bacterial microbiota of RTI clusters apart from that of the healthy control. Further, we used the UniFrac distance analysis to quantify the dissimilarity between bacterial communities among the groups (Fig. 3B). The distance between the BV group and healthy control was significantly increased compared to the distance between UU/CST I/CST II group and healthy control. This indicated that the bacterial communities in BV were distinct from healthy control, followed by UU. Additionally, the bacterial communities in CST II were most similar to healthy control, followed by CST I.

We used the Kruskal-Wallis to test whether there were significant differences in the relative abundance of *Lactobacillus* and pH in the three disease types and controls, and the Wilcoxon test was further used for pairwise comparison. Our results revealed that the median of *Lactobacillus* of VVC disease was closer to health than the other disease types and there were no significant differences between Health and VVC in pH (Fig. S2).

**Vaginal microbial alterations between different RTIs and healthy women.** Next, we performed linear discriminant effect size (LEfSe) to assess and distinguish the composition of the vaginal microbiome between the RTI and healthy control groups (17). We then evaluated the relationship between the biomarker microbiome and vaginal pH in different diagnosis types. In addition, we identified the biomarker microbiome that was significantly positively or negatively correlated with pH marked in red and green on the left side of the figure, respectively. LEfSe analysis revealed a significant increase in the relative abundance of different bacterial taxa in different groups. All potential biomarkers (LDA > 2) are shown in Fig. 4. For instance, the microbiome in the healthy control group was enriched by genera of *Lactobacillus*. The vaginal

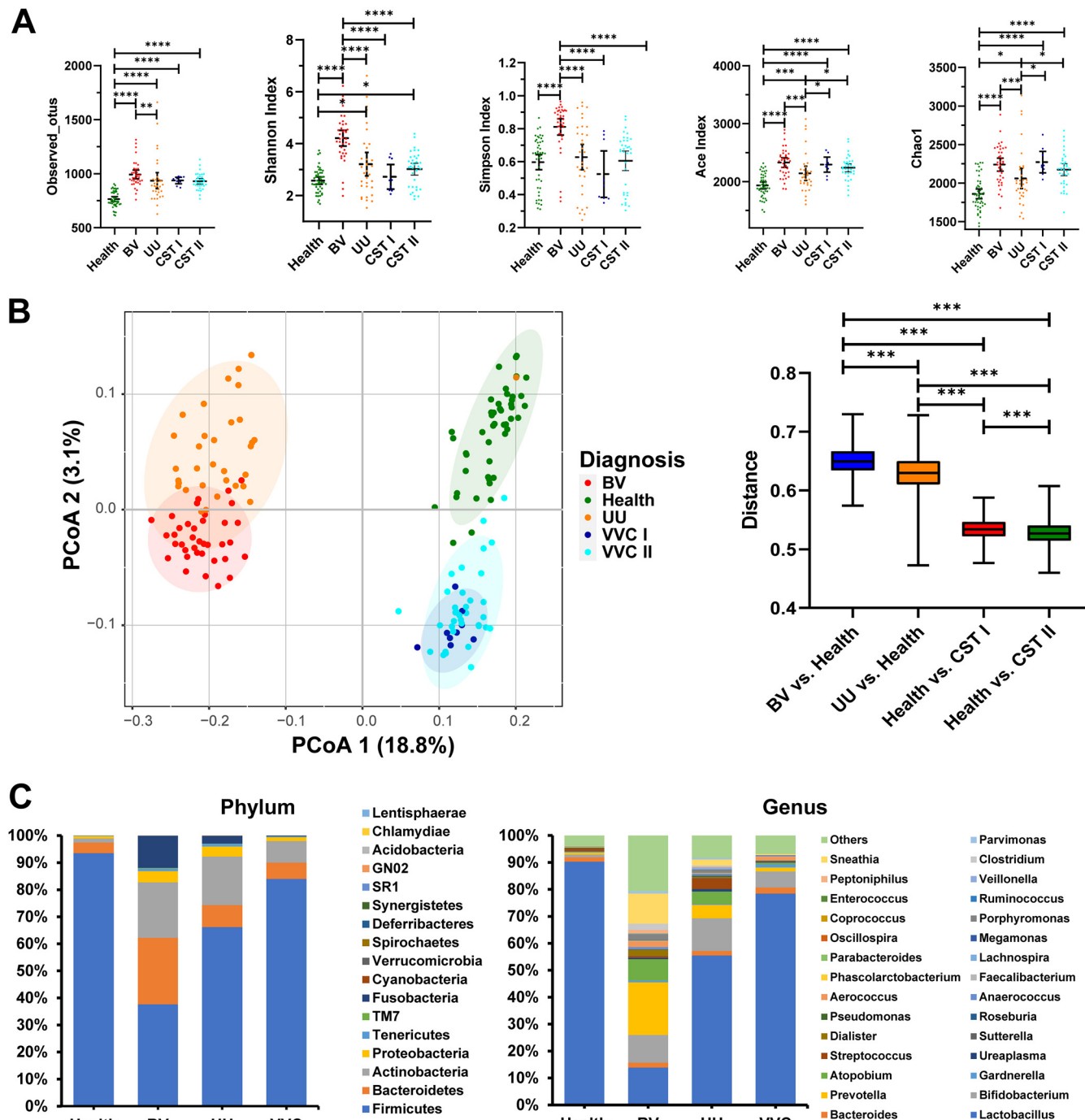

**FIG 3** Characteristics of microbial community composition in BV, VVC, UU and health control groups. (A) Comparison of $\alpha$ diversity of vaginal microbiota between three RTIs and health control groups. a-d, represent the Observed OTUs, Shannon, Ace, and Chao1 indexes, respectively. *, $P < 0.05$; **, $P < 0.01$. (B) PCoA of bacterial beta diversity based on the unweighted UniFrac distances between three RTIs and health control groups. (C) Relative abundance of different taxa at the phylum and genus levels between three RTIs and health control groups. BV, Women with bacterial vaginitis. VVC, Women with vulvovaginal candidiasis. UU, Women with vaginitis caused by *Ureaplasma urealyticum*.

microbiome of the BV group was characterized by a dominance of *Prevotella*, *Sneathia*, *Atopobium*, *Megasphaera*, *Shuttleworthia*, etc. Moreover, we found that *Shuttleworthia* was positively correlated with the increase of pH. The UU group was enriched by genus *Streptococcus*, *Anaerococcus*, and *Veillonella*. In VVC group, at the genus level, *Aerococcus* were more enriched in CST I, while *Gardnerella* and *Bacteroides* were more enriched in CST II ($P < 0.01$, Wilcoxon rank-sum test; LDA >2.0; Fig. 4).

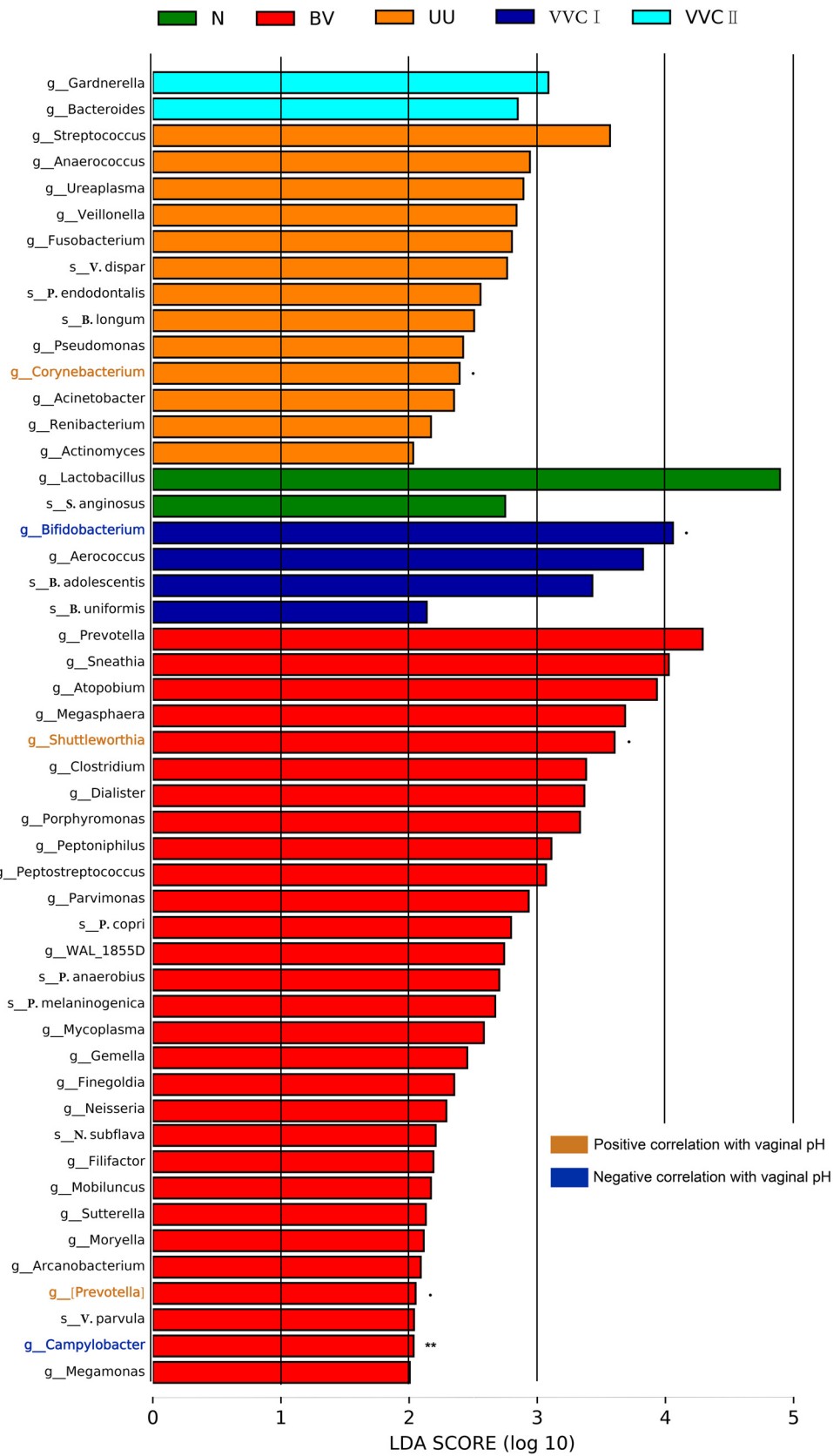

**FIG 4** Biomarkers of vaginal microbiome in patients with different RTI. Histogram of LDA scores. Computed for differentially abundant taxa between three RTIs and health control groups (LDA score above 2). The LDA score indicates the effect size and ranking of each differentially abundant taxon.

## DISCUSSION

Some discoveries have shown that bacteria and fungi could cooperate in a strategic evolutionary manner to affect some disease states (18). However, there is a dearth of information on the association of the vaginal microbiome with VVC pathogens. VVC has a profound effect on the quality of life for patients by virtue of the frequent recurrences (19). Asymptomatic colonization may persist for years in healthy women as yeasts live in symbiosis with vaginal microbiota, while this symbiosis will be broken when acute symptomatic occurred (20). Traditional clinical tests, including saline and KOH microscopy, have low sensitivity (21). However, with the increase of fungi that can cause VVC symptoms, especially *Candida glabrata*, it becomes critical to determine the species for effective treatment measures that ensure optimal management (22) and this requires a comprehensive understanding of the vaginal fungal environment.

The mycobiome of VVC patients vaginal were analyzed in this study. CST I and CST II were dominated by *C. glabrata* and *C. albicans*, respectively, two of the most common yeasts isolated in clinic (23). No significant differences in anaerobic, Gram-positive, potentially pathogenic microorganisms or pH values between CST I and CST II were found in our study, which may help explain VVC patients with two different fungal infections have similar clinical characteristics. In 44 specimens of VVC, mixed infections with NAC (Non-albicans Candida) species were observed, such as *Candida tropicalis* and *Clavispora lusitaniae*. *C. tropicalis* was reported emerged during the last 2 decades as new causative species of VVC (24). It is important to pay attention to other newly discovered fungi in VVC patients in clinic, and determining whether they are associated wit VVC merits further experimental validations.

Notably, the vaginal microbiome of VVC women exhibited distinct profiles between the two CSTs. *Candida* is the most prevalent fungal pathogen causing infections (25). The ability of this organism to infect and cause diseases is associated with biofilm formation, often involving interactions with bacteria on mucosal surfaces (26). Yet, most of the clinically used therapeutic approaches are monotherapies based on either antibacterial or antifungal agents despite the polymicrobial nature of disease-causing biofilms (15). Interestingly, we found that the pathogen of UU, *Ureaplasma*, was enriched in the CST II group. While *Prevotella*, a key signature in BV, was significantly increased in the CST I group. This may be associated with high incidence of mixed infection in VVC, which should be kept in mind for better diagnostic accuracy and treatment efficiency.

Compared with BV, UU and VVC, *Lactobacillus* dominated the vaginal microbiota of healthy women, which can produce glucose and lactic acid, resulting lower vaginal pH creates an unfavorable environment for the growth of pathogens. In addition, *Lactobacilli* may prevent the adherence of pathogenic microorganisms to vaginal epithelial cells through "competitive exclusion" and "bacterial interference" (27). Strikingly, the most similar vaginal bacteriome to that of healthy subjects was found in VVC patients' vaginas, with the abundance of *Lactobacillus* near to 80%. These results indicated that *Candida* infection could create an environment that was friendly to the growth of *Lactobacillus*, or maybe both are boosted by some environmental factor, like estrogen (28). The results also supported fungal infection was associated with intermediate microflora (a state between healthy vaginal microflora and BV vaginal microflora). In other words, the women with VVC may shift to other RTIs bacteriome (16). According to the literature, BV is a set of common clinical symptoms that can be provoked by a plethora of bacterial species with proinflammatory characteristics, which is characterized by a shift in the vaginal flora from the dominant Lactobacillus to a polymicrobial flora (29). The vaginal microbiome in BV is characterized by the highest diversity compared with other groups and altered composition at the phylum or genus level substantially. Similar to findings of previous studies, *Lactobacillus* are replaced with a high abundance of facultative and strict anaerobic bacteria in BV, including *Prevotella*, *Sneathia*, *Atopobium*, and other BV-associated bacteria (BVAB) (30). *Lactobacillus* is the main source of lactic acid that keeps the low pH value. One of the characteristics of BV is the decrease in relative abundance of *lactobacillus* in the vagina, which leads to high pH value (31). Another characteristic of BV is the increase of facultative anaerobic bacteria, such as *Prevotella*, *Sneathia,* and *Atopobium*. Previous

**TABLE 1** Characteristics of study participants[a]

| Variable | Health (n = 47) | BV (n = 34) | VVC (n = 44) | UU (n = 33) | P-value[b] |
|---|---|---|---|---|---|
| Age (yr) | 35.85 ± 8.207 | 39.84 ± 14.13[d] | 33.45 ± 8.382[d] | 36.36 ± 9.704[d] | 0.1855 |
| Vaginal pH (mean) | 4.302 ± 0.317 | 5.357 ± 0.15[c] | 4.345 ± 0.249[c] | 4.897 ± 0.375[d] | <0.0001 |
| VCD (I-II) | 47 (100%) | 0 | 0 | 0 | |
| VCD (III-IV) | 0 | 34 (100%) | 44 (100%) | 33 (100%) | |

[a]BV, Women with bacterial vaginitis. VVC, Women with vulvovaginal candidiasis. UU, Women with vaginitis caused by *Ureaplasma urealyticum*. VCD, vaginal cleaning degree.
[b]Kruskal-Wallis test. *P*-values adjusted by the method of Benjamini & Hochberg correction.
[c]Significance difference with Health (*P* < 0.0001).
[d]No significance difference with Health (*P* > 0.05).

research found that *Prevotella* was present in every case regardless of the Nugent score and it represented a high percentage of total species in BV group (32). *Prevotella* was associated with a positive whiff test, one of the clinical criteria comprising the Amsel test. *Prevotella* can produce polyamines during normal metabolic activity. These amines can increase the vaginal pH, which in turn may enhance the growth of other anaerobes associated with BV (33). In addition to *Prevotella*, another marker of BV, *Atopobium* is also reported associated with Amsel clinical criteria, including vaginal discharge and elevated pH. As for *Sneathia*, some authors concluded that it was epidemiologically associated with BV rather than being involved in the development of BV (29). Some bacterial genera that altered in BV, such as *Prevotella* and *Bifidobacterium*, showed a significant shift in UU as well, suggesting that the changes in vaginal ecology in UU were also featured by high abundance of anaerobic bacteria. *Prevotella*, *Corynebacterium* and *Shuttleworthia* were positively correlated with the increase of pH. The abundances of *Prevotella* and *Shuttleworthia* in the BV group increased significantly, which may be related to our finding of the highest pH value in the BV group. *Corynebacteriu*m, increasing in the UU group, can utilize glycogen stored in vaginal epithelial cells, causing a malodorous vaginal discharge characterized by an abnormally high pH (5.0 to 5.5) (34).

In conclusion, we integrated the analysis of mycobiome and bacteriome in VVC patients by next-generation sequencing. Mycobiome analysis determined two CSTs based on the dominant position of *Candida glabrata* or *Candida albicans* in the VVC group, and the vaginal bacteriome was also different in these two CSTs. The VVC group are is closer to the level of the healthy group in bacterial communities than other vaginal infections (BV and UU). Which is characterized by high relative abundance of *Lactobacillus*. We also found the vaginal microbiome positioning in UU group is between healthy group and BV-positive group. Both the infected groups were charactered by depletion of *Lactobacillus* and a corresponding increase in different facultative anaerobes (e.g., *Prevotella*, *Faecalibacterium*, and *Atopobium*).

## MATERIALS AND METHODS

**Sample collection and clinical information.** Healthy volunteers and patients with RTIs (VVC, BV, and UU) were recruited at the Dezhou People's Hospital, Shandong, China (Table 1). All subjects included in the study underwent routine gynecology examinations by two gynecologists. Forty-seven of the subjects were healthy, 44 were infected with VVC, 37 were infected with BV, and 33 were infected with UU. All patients were first diagnosed with an infectious disease, and mixed infections were excluded. VVC and UU were diagnosed via the microscopic detection of vaginal epithelial cells on a vaginal smear, physical examination, and the presence of secretions. VVC diagnosis was based on clinical symptoms (e.g., itching or vaginal whitish discharge), together with the microscopic and culture-based identification of Candida (35). The diagnosis of UU is based on analyzed with PCR for *Ureaplasma urealyticum* of vaginal swabs. The diagnosis of BV was established according to the guidelines Amsel's clinical criteria (36). All subjects were confirmed using Gram-stain criteria (Nugent scores) (37). Exclusion criteria were the following: less than 20 years of age, pregnancy, diabetes mellitus, use of antibiotics or vaginal antimicrobials in the previous month, menstruation, menoxenia, other reproductive tract infections, sex within 48 h, or diagnosed HPV or HIV infection. The study was approved by local institutional review boards (Dezhou People's Hospital, China), All recruited patients and healthy subjects provided written informed consent before sample donation and in compliance with national legislation and the Code of Ethical Principles for Medical Research Involving Human Subjects of the World Medical Association (Declaration of Helsinki).

**Sample collection and DNA extraction.** From each subject, we took two swabs after cleaning perineum and vulva, one from the vaginal fornix and the other one from the lower vagina (lower third of vagina). Whole swabs were collected in sterile tubes, immediately homogenized. Fresh samples were evaluated for

pH using pH-testing strips and then placed in a −80℃ refrigerator until DNA extraction. Microbial DNA was isolated from vaginal swab using the QIAamp DNA minikit (catnumber 51304, Qiagen) following the manufacturer's instructions. For each DNA sample, we amplified, respectively, the bacterial 16S rRNA genes using a primer set specific for V1–V2 hypervariable regions (27F 5'-AGAGTTTGATCMTGGCTCAG-3' and 355R 5'-GCTGCCTCCCGTAGGAGT-3'), and the internal transcribed spacer (ITS2) using a primer set specific (ITS3-2024F 5'-GCATCGATGAAGAACGCAGC-3' and ITS4-2409R: 5'-TCCTCCGCTTATTGATATGC-3') containing barcode sequences. The PCR products were checked using electrophoresis in 1% (wt/vol) agarose gels in TBE buffer (Tris, boric acid, EDTA) stained with Genecolour ITM (Gene-bio) and visualized under UV light. PCR products were pooled and purified using VAHTSTM DNA clean beads (Vazyme Biotech) according to the manufacturer's instructions. Amplicons were sequenced using the Illumina HiSeq platform with the 2 × 250bp paired-end protocol. Raw data were submitted to the National Omics Data Encyclopedia (NODE, https://www.biosino.org/node/index) with the accession number OEP002898.

**16S rRNA and ITS sequence analyses.** The paired-end data set was joined and quality filtered using FLASH (38). The 16S rRNA gene and ITS sequencing data were processed with QIIME V1.9.1 software package (39). Chimeric sequences were removed using usearch61 (40) with *de novo* models. UCLUST was used to subsequently cluster sequences that did not match any entries in this reference into *de novo* operational taxonomic units (OTUs) at 97% similarity. For both ITS and 16S rRNA gene data, a representative sequence was picked from each OTU, followed by its annotation or taxonomic assignment using reference databases GreenGenes_13.8 (for bacteria) and UNITE (for fungi). Finally, a total of 17,437 OTU for bacteria were retained, with the number of reads ranging from 44,667 to 49,627; and 657 OTU for fungi were retained, with the number of reads ranging from 28,978 to 34,940. The alpha and beta diversity were calculated from OTU tables using QIIME scripts. Alpha diversity was estimated by the Shannon, Ace, Chao 1 index and the number of observed OTUs. Beta diversity was measured by Unweighted Unifrac distance. Then the distance matrixes were performed through Principal Coordinates Analysis and the first two principal components were calculated by vegan package (version 2.5-7). QIIME scripts was used for analyzing similarities (Adonis) on beta diversity matrices, to determine significant differences among microbial communities. The significance of the Adonis test was assessed with 9,999 permutations. LEfSe (Linear discriminant analysis effect size) was run to determine enriched microbiome from each diagnosis type using relative abundances (17). Features of genus or species levels with Linear discriminant analysis (LDA) score > 2.0 and *P*-value < 0.05 were considered statistically significant. Linear regression models were used to describe the relationship between vaginal pH in and microbial features from LEfSe results in different diagnosis types, respectively. To predict organism-level microbial phenotypic characteristics using 16S rRNA gene sequence data, we implemented BugBase (https://bugbase.cs.umn.edu/) by defaults parameters.

**Statistical analyses.** The data were characterized using mean and standard deviation for continuous variable and constituent ratio for categorical variable. The Wilcoxon test and Kruskal–Wallis test were used to evaluate the difference between and among groups separately. Spearman's rank correlation was used to determine the statistical dependence between continuous variables. The false discovery rate (FDR) correction was used for multiple tests. Without special instructions, *P*-value < 0.05 were considered statistically significant. The R project based on 4.0.3. version.

**Ethical approval and consent to participate.** In this study, which was approved by the Institutional Review Boards of Qilu Hospital of Shandong University Dezhou Hospital & Dezhou People's Hospital (IRB number 2022008), sample collection began in July 2018. Written informed consent and questionnaire data sheets were obtained from all participants who visited the Qilu Hospital of Shandong University Dezhou Hospital & Dezhou People's Hospital and agreed to serve as sample donors, in compliance with national legislation and the Code of Ethical Principles for Medical Research Involving Human Subjects of the World Medical Association (Declaration of Helsinki).

**Data availability.** All sequencing data associated with this study were uploaded to the National Omics Data Encyclopedia (NODE, https://www.biosino.org/node/index) with the accession number OEP002898.

## SUPPLEMENTAL MATERIAL

Supplemental material is available online only.

**SUPPLEMENTAL FILE 1**, PDF file, 0.6 MB.

## ACKNOWLEDGMENTS

We are grateful to the participants who donated the biologic samples. We also want to acknowledge the laboratory manager at the Qilu Hospital of Shandong University Dezhou Hospital for planning part of the data collection.

We have no conflicts of interest to declare.

The study was supported by National Natural Science Foundation of China 82172320, TaiShan Industrial Experts Program tscy20190612, Shandong University Outstanding Young Scholars Program, and Shandong Provincial Key Research and Development Program 2018CXGC1219.

Lei Zhang and Guoping Zhao conceived and designed the project. Each author has contributed significantly to the submitted work. Ying Li, Jing Xu, and Wanhua Xue made the diagnosis clinically, designed and collected the clinical setting, underwent the ethical evaluation process, and performed informed consent and questionnaire data sheets with

patients. Changying Zhao, Ying Li, and Bin Chen collected clinical samples and patient information, consolidated and organized all data. Bin Chen and Kaile Yue analyzed the data. Changying Zhao and Zhenzhen Su drafted the manuscript. Ying Li, Bin Chen, Lei Zhang, and Guoping Zhao revised the manuscript. All authors read and approved the final manuscript.

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
