## [Reviewer comments · Microbiology Spectrum]

Microbiology Spectrum

Mycobiome study reveal different pathogens of vulvovaginal candidiasis shape characteristic vaginal bacteriome

Changying Zhao, Ying Li, Bin Chen, Kaile Yue, Zhenzhen Su, Jing Xu, Wanhua Xue, Guoping Zhao, and Lei Zhang

Corresponding Author(s): Lei Zhang, Shandong University

Review Timeline:

Submission Date:	September 3, 2022
Editorial Decision:	November 7, 2022
Revision Received:	December 3, 2022
Editorial Decision:	February 19, 2023
Revision Received:	February 27, 2023
Accepted:	March 1, 2023

Editor: John Osei Sekyere

Reviewer(s): The reviewers have opted to remain anonymous.

Transaction Report:

DOI: <https://doi.org/10.1128/spectrum.03152-22>

November 7, 2022

Prof. Lei Zhang
Shandong University
Jinan
China

Re: Spectrum03152-22 (Mycobiome study reveal different pathogens of vulvovaginal candidiasis shape characteristic vaginal bacteriome)

Dear Prof. Lei Zhang:

Link Not Available

Sincerely,

John Osei Sekyere

Journals Department
Reviewer comments:

Reviewer #1 (Comments for the Author):

The majority of the comments from my initial review (for mSphere) were addressed.

My preference would be to avoid the term "dysbiosis" and simply describe what you mean by dysbiosis (e.g. "low relative abundance of *Lactobacillus*" or "higher alpha diversity"). But if you choose to use the term "dysbiosis," then I'd recommend defining it in the text of the manuscript.

From my initial review, my question about fungi in the healthy, BV and UU groups was not addressed. I still think it would be interesting to assess fungal communities in healthy, BV and UU groups (in addition to the VVC groups represented in figure 1A)

but I don't think it is required for publication.

Reviewer #2 (Comments for the Author):

The conclusion should solely be dependent on the data obtained in this study. The current conclusion is not clear and should be revised accordingly.

The average age of the patients are different (almost statistically significant). The authors should explain and discuss possible consequences of this.

Why did the authors exclude patients with <20 y?

Why did the authors classify BV and UV in separate groups?

What is the distribution of bacteria in BV group? Differences in this very group is decisive for the data outcome. Please discuss.

In several sections the authors claim that there is a causative relation between the existence of *Candida* spp. and bacteria. In this study there is no evidence for that. If there are functional studies that were done this should be shown as well. All these should be replaced by association.

Staff Comments:

Preparing Revision Guidelines

Please return the manuscript within 60 days; if you cannot complete the modification within this time period, please contact me. If you do not wish to modify the manuscript and prefer to submit it to another journal, please notify me of your decision immediately so that the manuscript may be formally withdrawn from consideration by Microbiology Spectrum.

Dear editors and reviewers,

We sincerely appreciated your critical and insightful comments that helped us improve our manuscript.

We have carefully revised the manuscript according to reviewers' comments. According to the Reviewer#1's suggestions, we have modified the improper statement about the microbiome "dysbiosis". Furthermore, based on the suggestions of Reviewer#2, all the questions raised by the reviewer about Discussion and Conclusion that needed to be further clarified have also been addressed, and we have conducted a more careful proofread. In the manuscript, the revisions are highlighted in the Word Track Environment.

Below are our point-to-point responses to the reviewers' comments.

Responses to issues raised by reviewer 1:

1. The majority of the comments from my initial review (for mSphere) were addressed.

My preference would be to avoid the term "dysbiosis" and simply describe what you mean by dysbiosis (e.g. "low relative abundance of Lactobacillus" or "higher alpha diversity"). But if you choose to use the term "dysbiosis," then I'd recommend defining it in the text of the manuscript.

Response:

--Thank you for the kind reminder. We have realized the description of the term "dysbiosis" is improper and agreed with the reviewer that simple description is better than using the vague term "dysbiosis". Hence, we replaced the term "dysbiosis" by simple description in the revised version. Such as described it by "the relative abundance of microbes in vagin changed during VVC, such as *Lactobacillus* decreased, *Prevotella*, *Sneathia*, and *Atopobium* increased" . Relevant changes were

made in Lines 71-73

2. From my initial review, my question about fungi in the healthy, BV and UU groups was not addressed. I still think it would be interesting to assess fungal communities in healthy, BV and UU groups (in addition to the VVC groups represented in figure 1A) but I don't think it is required for publication.

Response:

--Thank you very much for raising this meaningful point. We greatly agree with you that it would be meaningful to assess fungal communities in healthy, BV and UU groups. The main aim of our study was to reveal the vaginal mycobiome and bacteriome in VVC patients. Meanwhile, according to previous report, only about 60% of asymptomatic women have fungal colonization in vaginal (Kalia et al., 2020, *Ann Clin Microbiol Antimicrob*). What's more, we performed microscopic examination of vaginal swabs during the diagnosis of patients with BV and UU, but no fungal spores or hyphae was found in these two groups. Some samples were also selected for PCR with universal fungal primers ITS in BV, UU and health groups but no positive bands were amplified. This maybe due to the limitations of our sample size and the lower likelihood of fungal colonization in these groups. Ultimately we did not assess fungal communities of BV, UU and health groups. We agree extremely with that it would be meaningful to assess fungal communities in these groups and expecting future studies to answer this question with larger sample sizes.

Responses to issues raised by reviewer 1:

1. The conclusion should solely be dependent on the data obtained in this study. The current conclusion is not clear and should be revised accordingly.

Response:

--Thank you for raising this critical point, we have carefully revised the conclusion to

highlight the important findings based on the data obtained in our study. In conclusion, we integrated analysis of mycobiome and bacteriome in VVC patients by next-generation sequencing. Mycobiome analysis determined two CSTs based on the dominant position of *Candida glabrata* or *Candida albicans* in VVC group, and the vaginal bacteriome was also different in these two CSTs. VVC group are closer to the level of the healthy group in bacterial communities than other vaginal infection (BV and UU). Which is characterized by high relative abundance of *Lactobacillus*. We also found the vaginal microbiome positioning in UU group is between healthy group and BV-positive group. Both the infected groups characterized by a depletion of *Lactobacillus* and a corresponding increase in different facultative anaerobe (e.g. *Prevotella*, *Faecalibacterium* and *Atopobium*). The detailed revision can be found in Line 283-292, Page 14-15.

2. *The average age of the patients are different (almost statistically significant). The authors should explain and discuss possible consequences of this.*

Response:

-- Thank you for raising this key point. We have checked this part of the question you raised and found our error. We sincerely apologize that the p-value between the four groups in Table 1 was wrong. After our careful examination, the correct p-value is 0.1855, which means there is no statistically difference in average age between the four groups, we have modified the p-value in Table 1.

In the design phase of the study, we tried to keep the age of the groups as non-significant as possible in order to avoid bias due to age differences. We used the Kruskal-Wallis test for multiple group comparisons, and the p-value was 0.1855. The Wilcoxon test results further show that there is no statistical difference in the age distribution of patients in any group and the control group (P value was adjusted by BH method for to reduce the probability of making a Type I error). The adjusted P values of the comparison between BV vs. health, VVC vs. health, UU vs. health, VVC

vs. BV, BV vs. UU and VVC vs. UU were 0.5449, 0.4155, 0.8842, 0.1986, 0.5449, and 0.4558. All of them were greater than 0.05. We apologize again and modified the table1 correspondingly.

3. Why did the authors exclude patients with <20 y?

Response:

--Thank you for your question, this is according to the Chinese legal marriageable age, the female needed to be older than 20 years. The vaginal procedures (including sampling of vaginal swabs) are not ethically acceptable before that age. What's more, the onset of VVC rarely occurs in younger age such as premenarchal period (Xie et al., 2017, *Cochrane Database Syst Rev*). To sum up, we excluded groups younger than 20 years old.

4. Why did the authors classify BV and UV in separate groups?

Response:

--Thank you for your question. BV and UU are considered two types infectious conditions of low pathogenicity in clinical (Vogel et al., 2006, *Acta Obstet Gynecol Scand*). BV is considered to be a vaginal Lactobacillus-depleted microbiota, and characterized by a shift in the vaginal flora from the dominant Lactobacillus to a polymicrobial flora (Delgado-Diaz et al., 2022, *Microbiome*). *Ureaplasma urealyticum*, similar with *Candida*, is a common commensal of the urogenital tract of sexually mature humans. It is considered as an important opportunistic pathogen. (Vogel et al., 2006, *Acta Obstet Gynecol Scand*). In order to better evaluate the bacteriome of VVC, we included two other common vaginal infectious (BV and UU) as a reference. We speculate that these two infections correspond to different changes in the vaginal microbiome. So, we classify BV and UU in separate groups.

5. What is the distribution of bacteria in BV group? Differences in this very group is

decisive for the data outcome. Please discuss.

Response:

--Thank you very much for raising such an important opinion. We apologize for not describing the distribution of bacteria in BV group clearly. The Figure 3 expressed the characteristics of the vaginal microbiome in BV groups, including the higher Alpha diversity and the unique relative abundance of vaginal microbes. In Figure 4, we showed markers of vaginal microbiome in patients with BV. According to your suggestion, we have redescribed the part of Discussion and we also summarized more details on the microbiome for BV group. For example, the characteristics of BV is *Lactobacillus* are replaced with some facultative anaerobic bacteria, such as *Prevotella*, *Sneathia* and *Atopobium*. These markers have been associated with some Amsel-clinical criteria, including vaginal discharge, elevated pH. The detailed revision can be found in Lines 260-273, Page 13-14.

6. In several sections the authors claim that there is a causative relation between the existence of Candida spp. and bacteria. In this study there is no evidence for that. If there are functional studies that were done this should be shown as well. All these should be replaced by association.

Response:

-- Thank you very much for your kind suggestion. As you mentioned, no functional studies were conducted in our research, so there is no evidence for causative relation between the existence of *Candida spp.* and bacteria in our study. We have removed all expressions about causative relation and revised them into "association" to express the relationship between fungi and bacteria more carefully. In future studies, we will perform functional validation of relevant mechanisms in the expectation of obtaining a causal relationship between fungi and bacteria.

Reference

Delgado-Diaz, D. J., et al. (2022). "Lactic acid from vaginal microbiota enhances cervicovaginal epithelial barrier integrity by promoting tight junction protein expression." Microbiome **10**(1): 141.

Kalia, N., et al. (2020). "Microbiota in vaginal health and pathogenesis of recurrent vulvovaginal infections: a critical review." Ann Clin Microbiol Antimicrob **19**(1): 5.

Vogel, I., et al. (2006). "The joint effect of vaginal *Ureaplasma urealyticum* and bacterial vaginosis on adverse pregnancy outcomes." Acta Obstet Gynecol Scand **85**(7): 778-785.

Xie, H. Y., et al. (2017). "Probiotics for vulvovaginal candidiasis in non-pregnant women." Cochrane Database Syst Rev **11**(11): CD010496.

February 19, 2023

Prof. Lei Zhang
Shandong University
Department of Biostatistics
Jinan
China

Re: Spectrum03152-22R1 (Mycobiome study reveal different pathogens of vulvovaginal candidiasis shape characteristic vaginal bacteriome)

Dear Prof. Lei Zhang:

Link Not Available

Sincerely,

John Osei Sekyere

Journals Department
Reviewer comments:

Reviewer #1 (Comments for the Author):

I appreciate that the authors addressed my preference for avoiding the term "dysbiosis."

It seems like many of the edits have introduced additional typos, grammar problems and confusing phrases. For example:

Lines 46-48: "VVC patients showed the vaginal bacteriome that is most similar to healthy subjects and the bacterial structure that is minimum difference to that of BV and UU." I think these lines are summarizing the results in Figure 3B - but the wording could be clearer.

Lines 255-256: "According to the literature, the clinical syndrome of BV is most likely caused by confusion or change of vaginal microbiota (29)." I am not sure what the authors mean by "...BV is most likely caused by confusion..." - maybe that there is confusion regarding the diagnosis of BV?

Select typos:

Line 73: "thses"

Line 529: "OUTs"

Staff Comments:

Preparing Revision Guidelines

Please return the manuscript within 60 days; if you cannot complete the modification within this time period, please contact me. If you do not wish to modify the manuscript and prefer to submit it to another journal, please notify me of your decision immediately so that the manuscript may be formally withdrawn from consideration by Microbiology Spectrum.

Dear editors and reviewers,

Thank you for taking time out of your busy schedule to review the manuscript. These suggestions have enabled us to improve our work. We're very sorry for the grammar problems and confusing phrases. We have taken Reviewer#1's suggestions seriously and have carefully corrected and replied the manuscript for this revision. The manuscript has been revised and re-polished. We hope it can meet the standard of Microbiology Spectrum now.

Below are our point-to-point responses to the reviewers' comments.

Responses to issues raised by reviewer 1:

1. Lines 46-48: "VVC patients showed the vaginal bacteriome that is most similar to healthy subjects and the bacterial structure that is minimum difference to that of BV and UU." I think these lines are summarizing the results in Figure 3B - but the wording could be clearer.

Response:

--Thank you for your careful reading of our manuscript. We apologize for any confusion caused and appreciate the valuable suggestions. We have replaced this sentence by " The vaginal bacteriome in VVC patients was between the healthy and other RTIs (BV and UU) status, it bore the greatest resemblance to that of healthy subjects. While BV and UU patients have the unique vaginal microbiota community structure, which very different with healthy women" to make it clearer.

Relevant changes were made in Page 3, Lines 46-49

2. Lines 255-256: "According to the literature, the clinical syndrome of BV is most likely caused by confusion or change of vaginal microbiota (29)." I am not sure what the authors mean by "...BV is most likely caused by confusion..." - maybe that there is

confusion regarding the diagnosis of BV?

Response:

---Thank you for raising this insight question. We're sorry for using this inaccurate description and have adjusted manuscript. We did not intend to indicate the confusion regarding the diagnosis of BV, and altered the sentence to specify that the symptoms of BV are often associated with vaginal microbiome fluctuations from the Lactobacillus-dominated status to another unhealthy structure. We changed this sentence into "According to the literature, BV is a set of common clinical symptoms that can be provoked by a plethora of bacterial species with proinflammatory characteristics, which is characterized by a shift in the vaginal flora from the dominant Lactobacillus to a polymicrobial flora".

Relevant changes were made in Page 13, Lines 257-260

3. Select typos:

Line 73: "thses"

Line 529: "OUTs"

Response:

--Thank you for your careful reading of our manuscript. We have changed "thses, OUTs" to "these, OTUs" (Page 5, Line 74 and Page 23, Line 529). We sincerely apologize for the inconvenience of reading caused by grammatical problems and typos. Accordingly, we have corrected the problems and all similar ones throughout the manuscript without altering the paper's original meaning. All revisions were highlighted in the new manuscript.

March 1, 2023

Prof. Lei Zhang
Shandong University
Department of Biostatistics
Jinan
China

Re: Spectrum03152-22R2 (Mycobiome study reveal different pathogens of vulvovaginal candidiasis shape characteristic vaginal bacteriome)

Dear Prof. Lei Zhang:

The manuscript still has several grammatical and language errors. Please get a better editor to revise it and remove all the language errors. If you don't do so, it can prolong the production of the manuscript or lead to its subsequent refusal.

Your manuscript has been accepted, and I am forwarding it to the ASM Journals Department for publication. You will be notified when your proofs are ready to be viewed.

Sincerely,

John Osei Sekyere
Editor, Microbiology Spectrum
